# Threat at One End of the Plant: What Travels to Inform the Other Parts?

**DOI:** 10.3390/ijms22063152

**Published:** 2021-03-19

**Authors:** Ralf Oelmüller

**Affiliations:** Department of Plant Physiology, Matthias Schleiden Institute of Genetics, Bioinformatics and Molecular Botany, Friedrich-Schiller-University, 07743 Jena, Germany; b7oera@uni-jena.de

**Keywords:** systemic signaling, calcium, reactive oxygen species, priming, salicylic acid, jasmonic acid, azelaic acid, glycerol-3-phosphate, dehydroabietinal, pipecolic acid, small RNA, miRNA, siRNA, vascular tissue, phloem transport, volatiles

## Abstract

Adaptation and response to environmental changes require dynamic and fast information distribution within the plant body. If one part of a plant is exposed to stress, attacked by other organisms or exposed to any other kind of threat, the information travels to neighboring organs and even neighboring plants and activates appropriate responses. The information flow is mediated by fast-traveling small metabolites, hormones, proteins/peptides, RNAs or volatiles. Electric and hydraulic waves also participate in signal propagation. The signaling molecules move from one cell to the neighboring cell, via the plasmodesmata, through the apoplast, within the vascular tissue or—as volatiles—through the air. A threat-specific response in a systemic tissue probably requires a combination of different traveling compounds. The propagating signals must travel over long distances and multiple barriers, and the signal intensity declines with increasing distance. This requires permanent amplification processes, feedback loops and cross-talks among the different traveling molecules and probably a short-term memory, to refresh the propagation process. Recent studies show that volatiles activate defense responses in systemic tissues but also play important roles in the maintenance of the propagation of traveling signals within the plant. The distal organs can respond immediately to the systemic signals or memorize the threat information and respond faster and stronger when they are exposed again to the same or even another threat. Transmission and storage of information is accompanied by loss of specificity about the threat that activated the process. I summarize our knowledge about the proposed long-distance traveling compounds and discuss their possible connections.

## 1. Introduction

Plants are permanently exposed to environmental changes, which can be beneficial or harmful. As sessile organisms, they cannot escape. In many cases, not the entire plant but only an organ, a few cells or a single cell of the plant is exposed to a particular stress or threat, which then activate an appropriate (defense) program. However, it is quite important that the neighboring cells or organs, distal parts of the plant or even neighboring plants are informed about the threat, so that they can respond to it as soon as the threat also reaches them. Systemic signaling is a typical plant response to stress. Herbivores, nematodes, insects, birds or pathogenic microorganisms attack a leaf, stem or a part of the root system. Abiotic stresses, such as UV light, drought, heat, wind or air pollution, can impair the aerial parts of a plant; other abiotic stresses, such as high salt concentrations in the soil, nutrient and water deficiencies, and heavy-metal or pesticide contaminations, are often first perceived by the roots. The distribution of the threat information requires signaling molecules, which either travel directly or activate processes which move from the stress-exposed organ away. Well investigated examples for traveling molecules are phytohormones, small metabolites, propagating electric, reactive oxygen species (ROS) and Ca^2+^ waves, as well as hydraulic signals. More recently, the importance of volatiles for the information spread became obvious. Many signaling molecules, such as hormones, can directly activate responses in the distal tissue; similar to those activated in the stress-exposed tissue. For instance, water stress induces abscisic acid (ABA), necrotrophic pathogens jasmonates and biotrophic pathogens salicylic acid (SA), and the downstream responses are the same in local and distant tissues. Other traveling chemical mediators, such as methyl-SA (MeSA), are linked to hormones, transport the information via the vascular system or air and are decoded in the distal tissue, e.g., by releasing the hormone from its conjugate. Long-distance transport of RNAs through the vascular tissue has the advantage that they can carry highly specific information, on the other hand, the plant has to determine which RNAs travel. The compounds establishing propagating electric, ROS and Ca^2+^ waves, or altered hydraulic pressure, do not carry specific information per se, and additional mechanisms are required for en- and decoding the specific information. The information can travel from cell to cell, apoplastically, and via the vascular tissue or even through the air as volatiles. How these signals are decoded in the neighboring not stress-exposed tissues and how they activate appropriate responses in distal tissue which are specific for the threat perceived at the other end of the plant is not really understood. In nature, the information spread is based on multiple, simultaneously occurring transduction events which are connected to each other. In order to activate an appropriate response, the distal tissue must receive sufficient information about the threat, which can be achieved by combinations of traveling compounds. The response in the distal tissue can range from a general stress response to a response that is a direct answer to the threat.

## 2. Priming and Information Storage

Beauverie [1] and Ray [2] described already in 1901 that plants show a stronger defense against a pathogen, when they are infected a second time. This implies that information about the first infection is somehow stored in the plant and that plants have a memory. Numerous groups have investigated this phenomenon called “priming” since its first discovery (see References [3,4,5,6,7]). A memory mechanism is important not only for the local response but also for responses in tissues which were not directly exposed to the stress.

## 3. Priming

Defense is energy consuming, and therefore a plant must decide whether it invests in growth or defense. It is long known that plants possess a constitutive and an inducible defense, and the latter one is only activated in response to an attack, in order to avoid excess investment into defense when there is no threat. Priming means that information about a previous attack is stored and re-activated upon a second attack, and this results in stronger and faster defense responses compared to those found in unprimed tissues. This is based on more efficient sensing of threat signals, a faster signal transduction (such as MAPK3/6 or WRKY transcription factor phosphorylation), leading to defense gene activation, but also faster synthesis/activation of defense-related phytohormones and defense-related secondary metabolites. A simple mechanism is the up-regulation of genes/proteins for the perception of the threat (receptor and co-receptors) and signaling compounds after the first stress stimulus, which might be inactive as long as no more stimuli are there. This has been shown for pattern recognition receptors [8] and MAPKs [9] in primed tissue. Priming resulted in higher pattern recognition protein levels [8] and the accumulation of mRNAs and inactive proteins of MAPK3/6 [9]. This elevated activation was linked to enhanced defense gene expression. The higher number of receptors/co-receptors/MAPKs in primed tissue allows a more efficient perception of pathogen-associated molecular patterns and signaling, as soon as a second attack occurs. Defense priming occurs locally and in distal tissue. There are also interesting reports demonstrating that plants which were challenged by a particular stress respond stronger to a different stress which happens later. This so-called “cross-tolerance” may occur when common signaling molecules are involved in the two stress responses and these signaling molecules can be maintained in an activated form between the two stimuli. Perez and Brown [10] discussed the role of ROS as a possible driver of cross-tolerance.

## 4. Long- and Short-Term Information Storage

Priming in a distal tissue means that the information that travels via volatiles, hormones, RNAs or other small chemical compounds must be stored, eventually over longer periods of time. Even transfer of the information to the next generation has been reported [11]. Information storage has been found to be associated with epigenetic changes at the DNA and small interference (si)RNA [12,13,14,15,16,17]. In eukaryotes, methylation and histone H3 and H4 acetylation controls transcription of many regulated genes over a long period of time. Thus, this type of chromatin modifications, among others, is likely involved in memorizing previously happened threat. Several studies showed that systemic acquired resistance (SAR) results in chromatin modifications, which prime systemic tissues for enhanced and rapid signaling [18].

Epigenetic changes are clearly important for long-term information storage, but in many cases, short time memory is required to allow efficient systemic signal propagation. Upon a local attack, the information travels within the plant body and the intensity of the response decreases with increase of the distance to the starting point. The propagation is often discussed in the context of feedback loops and cross-talks between traveling compounds/signals; however, an amplification of the traveling signal(s) could also be achieved if a cell within the propagation line could store the information and respond stronger to a new stimulus which arrived at the cell shortly after the first stimulus. This short-term storage is likely different from the long-term storage described above. It does not result in a stronger defense response, but primarily in the maintenance of signal propagation, i.e., electro-chemical mechanisms must maintain the cells in an “alarm situation”. It may last only for microseconds, or for a few minutes. Signaling molecules or channels, which are involved in information propagation must stay longer activated. A second wave of signals can now faster pass though this tissue and induce a stronger signal which might also travel longer distances. A promising model for such a short-time information storage is provided for the carnivorous plant Venus flytrap, *Dionaea muscipula*, which closes its leaves to capture insect prey. The closure response usually requires two successive mechanical stimuli to sensory hairs on the leaf blade within approximately 30 s. Thus, the first stimulus must be memorized and the information must be transduced from the sensory hair to the leaf blade. Suda et al. [19] showed that the signal memory is linked to Ca^2+^ dynamics. Stimulation of a sensory hair caused an increase in cytosolic Ca^2+^ concentration ([Ca^2+^]_cyt_) in the sensory hair and spreading to the leaf blade. A second stimulus increased [Ca^2+^]_cyt_ to an even higher level, meeting a threshold that is correlated to the leaf blade closure. Because [Ca^2+^]_cyt_ gradually decreased after the first stimulus, the [Ca^2+^]_cyt_ increase induced by the second stimulus was insufficient to meet the putative threshold for movement after about 30 s [19]. This study provides ideas of how propagating waves can maintain their intensities over longer times and distances, simply by a short-time memory effect that allows an amplification of the information along the traveling path. However, there are also other examples demonstrating that a second stimulus within a short period of time does not result in an additional response, for instance for stomata closure in response to ozone [20].

## 5. Traveling Ca^2+^, ROS and Electric Waves

Local infections induce a rapid increase in [Ca^2+^]_cyt_ levels, the establishment of an action potential across the plasma membrane (PM), and the generation of ROS in the apoplast via NADPH oxidases (Figure 1). These responses are connected and several studies have demonstrated that genetic inactivation of proteins required for establishment of a wave of one component eliminates also the waves of the two others (e.g., Reference [21]). However, how the responses are linked is not yet completely understood. For instance, it is technically possible to activate NADPH oxidase-dependent ROS production in the absence of Ca^2+^, and vice versa. The ROS responses occur not only in the local cell/tissue, but propagate to neighboring cells, including companion cells and sieve elements in the vascular system. The propagating Ca^2+^/ROS/electric wave is also crosslinked to the other systemic signaling compounds mentioned in this review, suggesting that a combination of different traveling compounds may allow specification of the threat information. Such a network may include nitric oxide (NO), hormones, osmotic pressure and the small metabolites discussed below, as well as volatiles. However, it is likely that individual members of the wave do not travel with the same speed, have different capabilities to pass through different tissues and reach different distances. How a link between the traveling compounds during signal propagation is established and maintained is still unknown.

## 6. ROS

Different ROS are produced in different cellular compartments. In lower concentrations, they function as signaling molecules for differentiation, growth, development, defense, control of metabolism or proliferation and activate appropriate signaling pathways. ROS-activated signaling events also lead to the activation of cell death and autophagy programs. In higher concentrations, ROS are dangerous and destructive for the cell; therefore, controlling the ROS levels in and around the cell is a major task for an organism (see References [22,23] and references therein). The role of ROS in systemic cell-to-cell signaling is part of a complex network which integrate ROS information with other signals. Central roles in systemic signaling play the PM-bound respiratory burst oxidase homologs (RBOH)D and -F. These NADPH oxidases generate superoxide radicals in the apoplast which are dismutated spontaneously or with the help of apoplastic superoxide dismutases to H_2_O_2_. H_2_O_2_ can enter the cell, e.g., via aquaporins (see Reference [24] and references therein). Activation of RBOHs in response to pathogens or microbe-associated molecular patterns occurs, among other mechanisms, through Ca^2+^ binding to its EF hand domains and/or by Ca^2+^-activated kinases which phosphorylate the membrane-bound enzyme at the N-terminal tail in the cytoplasm. *S*-nitrosylation also controls RBOH activity [25]. Recent studies also demonstrated that the C-terminus of RBOHD is a target for regulation by phosphorylation [26] and persulfidation [27]. Furthermore, Zhang et al. [28] identified the conserved MAPK4 SIK1 as required for pattern-triggered immunity in plants. SIK1 associates with, phosphorylates and stabilizes the central immune regulator BOTRYTIS-INDUCED KINASE1 (BIK1). The PM-associated cytosolic kinase BIK1 (and related proteins) acts downstream of multiple cell-surface immune receptors and coordinates multiple immune outputs. Upon perception of pathogens, SIK1 and activated BIK1 phosphorylate RBOHD to enhance ROS production and promote defense [28]. The elevated apoplastic ROS level produced by RBOHs around the activated cells stimulate “ROS enhanced ROS production”, which results in a wave of ROS production in neighboring cells (see details in Reference [22]). The propagation of the ROS wave is linked to a propagating Ca^2+^ wave and changes in membrane potential across the PM, i.e., electric signals, and has been observed in various cell types and tissues (Figure 1).

The propagation of [Ca^2+^]_cyt_ and electric waves requires the Ca^2+^-dependent glutamate-receptor like GLR3.3 and -3.6 channels (see Reference [29] and references therein; cf. below). Besides the link between Ca^2+^ and ROS through the Ca^2+^-binding EF-hand domains of the RBOHs, Ca^2+^-binding and activated signaling kinases or phosphatases are regulated by the two channels. How RBOHs are linked to electric waves, is not really understood yet. The electric potential across the PM may be established by the Ca^2+^ redistribution alone, or in combination with other ions, which require additional steps for their movement. Finally, changes in hydraulic pressure are often associated with traveling Ca^2+^/ROS/electric waves. It has been proposed that mechanosensitive channels could convert hydraulic wave information into Ca^2+^ signals [30].

## 7. ROS and Vascular Tissue

While initiated in stress-exposed cells, the propagating ROS wave ultimately reaches the vascular tissue. Zandalinas et al. [31] investigated the integration of ROS, Ca^2+^, electric, and hydraulic signals at the vascular bundles of *Arabidopsis*. Systemic signal propagation is impaired in mutants lacking RHOHD. Tissue-specific expression of *RBOHD* in phloem or xylem parenchyma cells of the *rbohD* mutant restores systemic ROS signaling to a local treatment of light stress. The authors also confirmed that, besides RBOHD, RBOHF is required for local and systemic ROS signaling at the vascular bundles. This study highlights the importance of RBOHD-driven ROS production at the vascular bundles, at least for light stress-induced systemic signaling.

Phloem and xylem parenchyma and bundle sheath cells appear to be involved in ROS signal propagation, but it is not clear whether they are bound to specific cell types in the vascular tissue (summarized in References [31,32,33]). For instance, the ROS wave is propagated together with a Ca^2+^ wave, and the latter depends on GLR3.3 and -3.6. *GLR3.3* is preferentially expressed in phloem cells whereas *GLR3.6* in xylem parenchyma cells (see References [29,33]). The authors also discuss the involvement of hormones in systemic signaling in the vascular tissue and propose that vascular bundles could serve as the central systemic signaling super highway. Light (phytochrome B) and hormone signaling are tightly linked and they have a tremendous influence on numerous processes in tissues which are far away from their perception/synthesis (e.g., Reference [34] and references therein). Hormone levels are ROS- [35] and Ca^2+^-regulated, and it is likely that traveling waves influence the hormone homeostasis to balance defense and growth responses.

## 8. ROS and Light/Phytochrome

Devireddy et al. [36] extended the studies on systemic ROS signaling and demonstrated that phytochrome B is required for systemic stomatal responses during light stress. The photoreceptor triggers the ROS wave under light stress conditions, which induces a systemic stress response in the systemic tissue and a memory mechanism that lasts for 3 to 6 h. Although operating at different time scales, the Venus flytrap model and the mechanism described above [19] provide examples for the involvement of memory mechanisms in systemic responses.

Fichman et al. [37] analyzed transcriptomic responses in systemic leaves and identified the transcriptional regulator MYB30 that orchestrates systemic ROS signaling and plant acclimation to high light stress. Interestingly, the *myb30* mutant shows an enhanced local response to light stress and enhanced ROS signal propagation, while 150 core transcripts associated with MYB30 function under light stress in the distal tissue failed to accumulate. This suggests that the lack of systemic acclimation drives a feedback mechanism with stimulates the local response and ROS signal propagation. Fichman et al. [37] concluded that the RBOHD-driven ROS wave triggers MYB30 in systemic tissue to induce the SAR response, and the decrease of the ROS signal. MYB30 function and regulation was interlinked with the GATA TRANSCRIPTION FACTOR8 and the GDSL esterase/lipase, At1g29670. Furthermore, the analysis of the expression profiles also linked MYB30 function in systemic signaling to the PHOTOCHROME-INTERACTING FACTOR4 (PIF4) suggesting an interaction between MYB30 and PIF4 in regulating the systemic ROS responses. These two investigations demonstrate that expression profiling may help to identify genes which are specifically expressed in systemic tissue in response to traveling signals. While the induced signaling events leading to altered expression patterns are probably the same in local and distal tissues, the activators appear to differ.

## 9. Ca^2+^

[Ca^2+^]_cyt_ accumulation requires receptors for regulating Ca^2+^ channels in response to the appropriate stimuli and decoding of the Ca^2+^ information involves Ca^2+^-binding proteins. In plants, Ca^2+^ binds and activates calmodulin, calmodulin-like proteins (CML), calcineurin B-like proteins (CBL), CBL-interacting protein kinases (CIPK) or Ca^2+^-dependent protein kinases (CDPK). It appears that regulation of the Ca^2+^-dependent responses is important for plants to distinguish between pathogenic and beneficial microbes: In general, pathogenic microbes induce cytoplasmic, while beneficial microbe nuclear Ca^2+^ elevations. The low [Ca^2+^]_cyt_ concentration increases upon a stimulus from the much higher apoplastic Ca^2+^ pool in the cell walls or from internal stores where the Ca^2+^ concentrations can be up to millimolar concentrations [38] (Figure 1).

The requirement for local Ca^2+^ changes in response to stimuli has been described extensively and one of the best studied stimuli for [Ca^2+^]_cyt_ elevation is flg22, a conserved peptide from the flagellum of different *Pseudomonas* species. After recognition by the FLAGELIN-SENSING2 (FLS2) receptor kinase in the PM of the host, the receptor dimerizes with BRASSINOSTEROID INSENSITIVE1-ASSOCIATED KINASE1 (BAK1), which results in the downstream activation of BIK1. It is believed that other stimuli, such as chitin or cello-oligomers, trigger different Ca^2+^ responses, i.e., that the concentration, location and temporal distribution of Ca^2+^ within the cell differ. Thus, different stimuli induce different Ca^2+^ signatures. They are deciphered on the basis of their spatial and temporal features, whereas different Ca^2+^ signatures determine the specificity of the responses.

### 9.1. Ca^2+^ Transport

Most of the studies performed so far revealed that Ca^2+^ is taken up from the apoplast. However, cell-internal stores also participate in regulating [Ca^2+^]_cyt_ concentrations. A number of Ca^2+^ channels have been investigated: The Cyclic Nucleotide-Gated Channel (GNGC) family with 20 members in *Arabidopsis* is involved in developmental and stress responses, as well as immunity. They have partially overlapping functions, form heterotetrameric complexes, and many of them appear to be localized to the PM (see Reference [39], for details). The CNGC2 and CNGC4 proteins in *Arabidopsis* together—but neither alone—assemble into a functional Ca^2+^ channel that is blocked by calmodulin in the resting state. Upon pathogen attack, the channel is phosphorylated and activated by BIK1 and this triggers [Ca^2+^]_cyt_ elevation [40]. The Ca^2+^ channel CNGC19 regulates *Arabidopsis* defense against *Spodoptera* herbivory [41]. JA induces cAMP accumulation, which activates the CNGC2 channel as an activating ligand, and promotes Ca^2+^ influx from the apoplast [42]. Examples for the involvement of Glutamate-Receptor-Like Channels (GLRs) (also 20 members in *Arabidopsis*) in systemic signaling propagation are discussed in Grenzi et al. [43]. Propagation of [Ca^2+^]_cyt_ and electric waves depends on GLR3.3 and -3.6 (see Reference [29] and references therein). Furthermore, the auto-inhibited calcium ATPases (ACAs) and CAX (Ca^2+^/proton exchanger) family members participate in establishing Ca^2+^ signatures and are also found in different cellular membranes. ACA pumps expel Ca^2+^ from the cytosol [44] (cf. below). Finally, Thor et al. [45] showed that BIK1 regulates plant stomatal immunity by interacting with and phosphorylating the N-terminal cytoplasmic loop of the recently discovered Ca^2+^ channel OSCA1.3 (At1g11960). As mentioned above, BIK1 phosphorylates also RBOHD in response to perception of pathogen/damage-associated molecular patterns, and the same phosphorylation motif is present in OSCA1.3. This links BIK activation to OSCA1.3-dependent Ca^2+^ fluxes and RBOHD-mediated ROS production.

Hilleary et al. [46] demonstrated that also tonoplast-localized Ca^2+^ pumps regulate Ca^2+^ signals during pattern-triggered immunity in *Arabidopsis*. The authors used flg22 to trigger immune responses in mutants missing tonoplast Ca^2+^ pumps. Besides uptake of Ca^2+^, the ACA pumps expel Ca^2+^ from the cytosol. The two tonoplast-localized pumps, ACA4 and ACA11, impact flg22-dependent Ca^2+^ signaling and related defense responses. The double-knockout *aca4/11* exhibited increased basal Ca^2+^ levels and Ca^2+^ signals of higher amplitude than wild-type plants. Relocalization of ACA8 from its normal cellular locale of the PM to the tonoplast also suppressed the *aca4/11* phenotypes [46]. Therefore, regulation of vacuolar Ca^2+^ sequestration is an integral component of plant immune signaling. Furthermore, the action of tonoplast-localized Ca^2+^ pumps does not require specific regulatory elements not found in PM-localized pumps [46]. Apparently, Ca^2+^ from different pools and Ca^2+^ pumps located in different cellular membranes contribute to plant immune responses. A stimulus may activate a combination of PM- and tonoplast-localized im- and exporters which induce a stimulus-specific Ca^2+^ signature in the local cell/tissue. It remains to be determined whether these Ca^2+^ signatures are maintained during traveling.

Well-investigated propagating Ca^2+^ waves are induced by herbivores or mechanical wounding (e.g., Reference [47]). These waves propagate with a speed of approximately 1 mm/s, and they are faster than the mass flow. The traveling Ca^2+^ wave is accompanied by an electric signal, namely slow wave potentials, a ROS wave, hydraulic signals and often result in downstream jasmonate responses. Whether the established Ca^2+^ gradient across the membranes alone establish and maintain propagating slow wave potentials, or whether other ions and compounds are also involved in it, remains to be determined. Kumari et al. [48] postulated the involvement of the H^+^-ATPase 1. Since propagating Ca^2+^ and electric waves require GLR3.3 and -3.6, one possible explanation for their involvement is the accumulating glutamate at the infection site. Furthermore, extracellular pH changes appear to be involved in the signal propagation. Since deletion of one of the postulated compounds prevents also traveling of the other compounds, it is difficult to clarify the exact role of each of the contributors to signal propagation. Again, it is also unclear whether the regulatory circuits, which are required for a local response to a particular stimulus are identical to those which mediate signal propagation. Furthermore, it is not understood whether traveling compounds induce a Ca^2+^-dependent response in distal tissues but are not involved in the local response.

### 9.2. Ca^2+^ Changes during Beneficial Symbiosis

This becomes even more complicated considering Ca^2+^-dependent systemic responses induced by beneficial fungi and bacteria, such as mycorrhiza and rhizobacteria in the roots. In contrast to defense responses, most of the Ca^2+^ responses induced by these beneficial microorganisms alter Ca^2+^ levels and affect Ca^2+^-dependent signaling compounds in the nucleus [38]. Nuclear ion channels, including DMI1 (DOES NOT MAKE INFECTIONS1) and the homologs CASTOR and POLLUX, are essential for the establishment of the symbiosis of legume with their root-colonizing microbes by generating nuclear and perinuclear Ca^2+^ spiking. Whether these Ca^2+^ responses are connected to those in the cytoplasm and whether systemic responses induced by beneficial microorganisms involve propagating Ca^2+^ signals, is not known. Although the Ca^2+^ signatures induced by beneficial and pathogenic microbes clearly differ, connections between them are likely.

### 9.3. Ca^2+^ Measurements

Changes in intracellular Ca^2+^ levels are often measured with the aequorin technology, a Ca^2+^-binding protein which emits light after activation. The outcome of this detection system is a rapid increase in [Ca^2+^]_cyt_ levels within minutes or seconds after the application of a stimulus followed by a rapid decline. This requires a synchronized activation of Ca^2+^ importers and exporters. CNGCs and GLRs either alone or in various combinations mediate the Ca^2+^ influx, whereas ACA8, -10, -12 and -13 are candidates for the Ca^2+^ efflux (cf. above). Cell-type-specific Ca^2+^ changes can be more accurately measured with the R-GECO system. It detects oscillations in the Ca^2+^ levels up to 30 min and the oscillation pattern is dependent on the stimulus, but also environmental factors (see Reference [38]). In molecular terms, signaling compounds and Ca^2+^ transporters responsible for Ca^2+^ changes can be identified with the aequorin technology and appropriate mutants; however, the higher cellular resolution of the Ca^2+^ responses with the R-GECO system demonstrates that the Ca^2+^ responses are more complex. The different oscillations patterns are regulated at the protein activity levels. Again, it remains to be determined whether the local Ca^2+^ oscillations are maintained during the propagation of the information to distal tissues.

## 10. Hydraulic Signals, Mechanosensitive Channels and Peptides

It has been long known that the regulation of cell expansion is controlled by the water status and, thus, hydraulic signals. Water is transported from the roots to the aerial parts and pressure changes are directly measurable in xylem vessels. These changes propagate to the surrounding cells and tissues but cannot be directly perceived by the dead xylem cells. The hydraulic signal propagates through the vasculature tissue with a proposed speed of 1 m min^−1^ (see Reference [49]). Ultimately, the turgor changes can be translated into physiological responses after the activation of mechanosensitive channels in the PM. Three families of plant mechanosensitive ion channels have been characterized: the MscS-like (MSL), Mid1-complementing activity (MCA) and two-pore potassium (TPK) families. They are found in multiple cellular compartments, and conduct Cl^−^, Ca^2+^ and/or K^+^ ions [50]. However, how this ion changes are connected to other traveling information or participate in establishing the electric wave is less investigated. A good example is an organ exposed to water shortage. The osmotic changes affect neighboring cells and activates pathways leading to ABA production, which stimulates downstream osmotic stress responses, including stomata closure (reviewed in Reference [51]). The key enzyme in ABA biosynthesis, NINE CIS EPOXYCAROTENOID DIOXYGENASE3 (NCED3), is highly expressed in the vasculature [52]. Under water-deficiency conditions, the CLAVATA3/EMBRYO-SURROUNDING REGION-RELATED25 (CLE25) peptide travels from the roots to the leaves, to activate ABA biosynthesis by stimulating *NCED3* expression [53,54,55,56,57]. CLE25 perception in the leaves occurs through the receptor-like kinases BARELY ANY MERISTEM (BAM)1 and BAM3. This is a nice example for the involvement of a peptide in long-distance ABA signaling through the vascular tissue (see Reference [58]).

Long-distance peptide signaling is also essential for nutrient homeostasis in plants [59], but much less is known about their involvement in plant immunity. The mechanisms and functions of root-to-shoot long-distance peptide signaling may provide examples for future studies in immune responses and the identified peptides could also be involved in other long-distances responses (see Reference [59]). The ABA example provided above also emphasizes the connection between long-distance peptide and hormone signaling.

## 11. Phytohormones Involved in Biotic Defense Responses

The involvement of phytohormones in systemic information distribution is not surprising, since hormones per se are defined as chemical mediators which induces responses in neighboring or distal tissues. I focus on SA and JA, which can travel through the vascular tissue and are involved in biotic stress responses.

### 11.1. SA

The SAR, a systemic response of the entire plant to a local infection is connected with SA. SAR function is coupled to SA accumulation in systemic tissues, whereas the hormone activates NONEXPRESSER OF PATHOGENESIS-RELATED GENES1 (NPR1) through a redox-regulated oligomer-to-monomer reaction in the cytoplasm, which is followed by defense gene activation in the nucleus [60]. This was observed in local and distal tissues; however, SA biosynthesis during SAR is much less stimulated in distal tissues. It became clear that additional factors participate in the SA-mediated SAR responses. MeSA was identified as a traveling signal (cf. below); however, also several compounds not directly related to SA participate in SA responses in systemic tissues.

Breitenbach et al. [61] identified the apoplastic LECTIN-LIKE PROTEIN1 (LLP1) and showed that it is required for systemic signaling. LLP1 is a member of a huge carbohydrate-binding protein family in *Arabidopsis* and overexpressor lines did not trigger a significant resistance response. Moreover, *llp1* knockout lines were not affected in their local resistance to *Pst/AvrRpm1*, while SAR was completely abolished in the mutants. The authors showed that LLP1 is linked to systemic rather than local immunity and suggest that LLP1 may act in parallel with SA.

Bernsdorff et al. [62] showed that the lysine catabolite pipecolic acid (Pip) participates in SAR but activates a distinct set of defense-related genes than SA (cf. below). Furthermore, Lee et al. [63] demonstrated that SA-independent systemic signals induce a gene encoding SNF1-RELATED PROTEIN KINASE 2.8 (SnRK2.8), which phosphorylates NPR1 during SAR. The SnRK2.8-mediated phosphorylation of NPR1 is necessary for its nuclear import. Although *SnRK2.8* transcription and SnRK2.8 activation are independent of SA signaling, the SnRK2.8-mediated induction of SAR requires SA. Together with the SA-mediated monomerization of NPR1, these observations indicate that SA signals and SnRK2.8-mediated phosphorylation coordinately function to activate NPR1 via a dual-step process in developing systemic immunity in *Arabidopsis* [63]. These examples show that SA-independent compounds either travel or are at least required for the establishing of the SAR response in the distal tissue.

The SAR is probably the best investigated mechanism for traveling compounds though the plant body, mainly the vascular tissue. Recent progress in the SAR research clearly demonstrates the connections to other systemic signaling molecules.

### 11.2. JA

JA is an oxylipin phytohormone from the oxygenation of polyunsaturated fatty acids [64]. The active from, jasmonoyl-L-isoleucine (JA-Ile), is bound to the Skp1-Cullin1-F-box-type (SCF) ubiquitin ligase complex SCF^COI1−JAZ^, which then degrades jasmonate ZIM domain (JAZ) transcription repressors [65] and initiates the expression of JA-responsive genes. How this response is activated in distal not threat-exposed tissue is still not clear [66,67,68,69,70]. Similar to systemic SA signaling, the current knowledge suggests an involvement of JA-dependent and -independent mobile signals [71,72]. JA-independent signals could be propagating waves or volatiles. Moreover, GLR3.3/6-dependent Ca^2+^ and electric waves have also been proposed to activate a lipoxygenase for JA/-JA-Ile biosynthesis and/or inactivate JAV1, a repressor of JA biosynthesis [73,74]. Evidence for JA-dependent systemic JA signaling comes from grafting experiments combined with feeding of labeled jasmonate forms. Schulze et al. [75] showed that *cis*-12-oxo-phytodienoic acid (OPDA), the precursor of JA, but not JA-Ile, travels from wounded shoots to roots through the phloem. Other studies suggest that JA and/or JA-Ile could be mobile (see Reference [72] and references therein). Since JA is accumulating in the phloem sap, in vicinity to tissues exposed to JA-inducing threats, it needs to be uploaded into the phloem. Li et al. [76,77] identified the jasmonate transporter (JAT1), as well as JAR3/4 (JA RESISTANT3/4), that mediates the cellular export of JA into the phloem. They conclude that cell-to-cell transport drives the loading and leaf-to-leaf translocation of JA, in a self-propagating manner. Furthermore, JA biosynthetic enzymes and signaling components are expressed in the core phloem cells and their function is amplified via feedback regulation [78,79]. Therefore, this auto-amplification mechanism could transmit e.g. wound signals along the phloem [80,81]. To what extend a systemic jasmonate response is induced by traveling jasmonate molecules moving long distances or whether the information is transmitted by a permanent de novo synthesis of the jasmonates along the traveling path, is still not resolved. Furthermore, genetic evidence for links to other traveling signals, such as Ca^2+^ and electric waves, volatiles or the small chemical compounds involved in systemic signaling (cf. below) in the phloem, is weak (see Reference [82]).

SA and JA have antagonistic functions and inhibit each other at several levels. Therefore, how systemic signaling induced by the two hormones is coordinated is not investigated.

## 12. Small Chemical Compounds Which Travel

The literature contains several chemical compounds involved in systemic signaling that have been long studied in SAR. The major players are the SA derivative MeSA, the plastid lipid-derived oxylipin azelaic acid (AzA), glycerol-3-phosphate (G3P), dehydroabietinal (DA), the lipid transfer protein DIR1 (DEFECTIVE IN INDUCED RESISTANCE1), the lysine catabolite Pip and its derivative N-hydroxy-Pip (NHP), volatiles like pinene (cf. below) or NO and extracellular (e)NAD(P) or eATP [83,84,85]. This list already shows the complexity in the signal propagation process: hormones (SA), volatiles and gases (MeSA, NO and pinene), lipid-associated compound (G3P), and energy carrier (NADP or ATP) are involved in signal propagation (Figure 2).

### 12.1. SA Derivative MeSA

MeSA is as a phloem-mobile compound [86]. It is synthesized from SA in tissue exposed to pathogens by different SA methyltransferases, and the accumulating SA levels in the infected tissue also inhibit the activity of the MeSA esterase, which catalyzes the conversion of MeSA into SA. This results in an even higher MeSA level. When MeSA reaches the systemic tissue, it is converted again to SA to induce the SAR response (Reference [86] and references therein). Although MeSA is transported through the phloem, the compound is highly volatile. This broadens its possibility to function as a long-distance signaling compound which does not only reach distal tissues of the infected plant but can participate in the propagation of the defense information within the plant community. The universal nature of MeSA also suggests that it functions in different plant species and has the potential to inform the entire community about a threat. Recently, Chen et al. [87] demonstrated that MeSA glucosylation by UGT71C3 facilitates negative regulation of the SAR response by modulating homeostasis of MeSA and SA. After primary pathogen infection, *ugt71c3* knockout mutants exhibited more powerful local and systemic resistance to secondary pathogen infection than wild-type plants, whereas the resistance response in *UGT71C3* overexpression lines was compromised [87]. Although volatile, the MeSA homeostasis within the plant is important for fine-tuning the SAR response. This study highlights the role of glucosylation of MeSA and potentially other systemic signals in negatively modulating plant systemic defense.

However, Adam et al. [88] also summarized arguments that question an important role of MeSA in systemic signal propagation. Examples include the following: (i) The S-adenosyl-L-methionine-dependent methyltransferase1 (*bsmt1*) mutants could induce a SAR response; however, they could not accumulate and evaporate MeSA in/from the primary inoculated leaves. (ii) The volatile concentration of MeSA (10–1000 µg L^−1^) required for the induction of resistance is two orders of magnitude higher than the concentration under normal conditions. (iii) *ISOCHORISMATE SNYTHASE1* gene expression and de novo SA biosynthesis are required for systemic SA accumulation, both in wild-type and *bsmt1* mutants, suggesting that airborne MeSA and/or MeSA transported via the phloem system alone may not be the SA source in systemic tissues [89].

### 12.2. Azelaic Acid (AzA)

Lipids range from small lipophilic molecules like the dicarboxylic acid AzA to complex phosphoglycerolipids and are involved in development, as well as various stress responses. The C_18_ fatty acids have a double bond between C9 and C10, which can be cleaved to the C9 compound 9-oxononanoic acid that is potentially converted to AzA. AzA was originally identified as a metabolite that accumulates after *P. syringae* infection of *Arabidopsis* leaves [90]. The hydrophobic oxylipin AzA is synthesized in plastid membranes, similar to the early steps in JA biosynthesis, and was identified as a systemic signal that induces defense responses in the presence of SA. When exogenously applied to leaves, it induces a primed state which is similar to the SAR after pathogen infection [91]. When applied to roots, it also induced resistance against *P. syringae* infection in leaves, and inhibits root growth [91]. The priming effects of AzA in systemic tissues require SA, since application of AzA alone is not effective [90]. Jung et al. [90] also showed that AzA accumulates in the vascular tissue of *Arabidopsis* upon *P. syringae* infection. Lim et al. [92] proposed that AzA and G3P (cf. below) is transported via the symplastic route and the plasmodesmata (PD) to the phloem. The PD-localized proteins PDLP1 and -5 (cf. below) regulate passage through the PD and control the symplastic transport of AzA (as well as other systemic signaling molecules such as G3P and mRNAs). In contrast, SA moves via the extracytosolic apoplast compartment. How far AzA travels is not well known. Recent studies by Cecchini et al. [91] showed that deuterium-labeled AZA applied to the roots does not move to aerial tissues; however AzA is normally found in roots and leaves. This suggests that AzA moves preferentially in its environment, and that the AzA effects in distal tissues may involve one or more additional mobile signals [91], which induce de novo AzA biosynthesis in the distal tissues. How this is synchronized with SA, is still an open question.

The effects of AzA on root growth, root-to-shoot signaling and disease resistance depend on the two lipid transfer proteins (LTPs) AZELAIC ACID INDUCED1 (AZI1) and EARLY ARABIDOPSIS ALUMINUM INDUCED1 (EARLI1), since the *azi1* and *earli1* knockout lines show reduced AzA and SAR effects. Interestingly, AZI1 is also required for other proposed systemic and interplant defense signals (G3P, DA and pinene/monoterpenes) (see References [91,93,94] and references therein). This clearly shows a link between the signaling compounds. Several studies position AzA downstream of ROS and upstream of G3P during systemic signaling (see Reference [95]), since AzA induces genes involved in G3P biosynthesis (see also Reference [96] and references therein). Furthermore, Yu et al. [93] proposed an intricate feedback regulatory loop among G3P, DIR1 (cf. below), and AZI1. Although these studies show dependencies between the signaling compounds, it is not clear how the information reaches the systemic tissue. AzA induces *azi1* gene expression and AZI1 can be phosphorylated by MAPK3/6, at least in vitro [97]. This suggests that AzA application triggers immunity through an AZI1/EARLI1/MPK3/6-dependent pathway.

In leaves AZI1 and EARLI1 are found in the ER, PM, PD and outer envelope membrane of the plastids [98]. More recently, Cecchini et al. [99] also showed that a pool of AZI1 exists at the plastid envelope, the site of AZA production. After a stimulus, the proportion of AZI1 located to plastids increased. Moreover, MAPK3/6 promote the accumulation of AZI1 at plastids during priming induction. This suggests the existence of a mechanism that links plastid targeting to defense responses [99].

Considering cell-to-cell movement, Cecchini et al. [98] proposed that LTPs participate in the formation of membrane contact site complexes which allow the movement of lipidic molecules through non-vascular transport. The complexes could play essential roles in the movement of lipid systemic signals to systemic tissues. Singh et al. [96] showed that AzA induces G3P biosynthesis and both metabolites are transported through the PD. PD permeability is regulated by PDLP1 and -5; however, these proteins also contribute to the partitioning of AZI1 between the plastids and cytoplasm. In the cytoplasm they might associate with PDLPs and, thus, influence PD gating. This might not only be important for AzA and G3P movement, but also other compounds involved in systemic distribution of information (e.g., RNAs; cf. below).

Besides the direct involvement in systemic signaling, AzA has been repeatedly identified in other studies. For example, the spent culture supernatant of *P. syringae* contains AzA [100] which raises the question of its role in plant–microbe interactions and the microbial effects on host immunity [7]. Furthermore, analysis of systemic root metabolome points to glucosylated AzA as potential microbe-induced signaling molecule that is subsequently exuded as free AzA [101]. It appears that AzA has a quite broad spectrum of functions, and has also been considered as a general marker for lipid peroxidation [102].

### 12.3. Glycerol-3-Phosphate (G3P)

G3P is a component of glycolysis and glycerolipid biosyntheses. The steady state level of G3P in a cell is established by biosynthesis and catabolism multifamily enzymes with different cellular locations. SAR induction requires G3P (one branch) and SA (a parallel branch) [103], and the free radicals NO and ROS are upstream activators of G3P (and AzA) functions. As described for AzA, SA and G3P accumulate in the infected leaves, but only a small portion of them is transported to distal uninfected leaves. As mentioned above, phloem loading of SA occurs via the apoplastic route, whereas that of G3P (as well as of AzA) via the symplastic route. PDLP1 and PDLP5 regulate PM gating, as well as the subcellular partitioning of G3P. The requirement of several compounds for a systemic response which utilize different routes for propagation opens the question how a synchronized response can be achieved in the distal tissue. Since only a small proportion of SA, G3P (and AzA) is transported, the majority of these compounds in the distal tissue must be synthesized de novo.

Shine et al. [104] studied rhizobia-induced systemic signaling in soybean. They showed that root recognition of incompatible rhizobia generates unknown signal(s), which travels to the shoot to induce the accumulation of G3P for systemic defense responses. G3P and SA in the shoots are required for Pip biosynthesis, which initiates a regulatory loop of NO/ROS–AzA–G3P accumulation. Transport of G3P back to the root enables root exclusion of incompatible rhizobia [104]. Knockdown mutants of G3P biosynthesis demonstrated overlaps with SAR signaling since they are compromised in both rhizobia- and pathogen-induced systemic resistance [104].

A possible connection between a ROS/Ca^2+^/electric wave and G3P and AzA for propagation in the vascular tissue needs to be investigated. G3P and AzA are induced by ROS and the signaling molecules travel in the vascular tissue (Figure 2).

### 12.4. Pip and Its Derivative N-Hydroxy-Pip (NHP)

NHP is a promising chemical mediator for the propagation of systemic defense signals and the knowledge of its role as traveling compound has strongly increased in recently years (summarized in Reference [105]). NHP biosynthesis occurs in three steps [106,107]: The plastid-localized enzyme AGD2-LIKE DEFENSE RESPONSE PROTEIN 1 (ALD1) converts lysine to ultimately 1-piperideine-2-carboxylic acid (P2C), and the reductase SAR-DEFICIENT4 (SARD4) catalyzes the reduction of P2C to Pip, which, after export form the chloroplast, is converted to NHP by the FLAVIN-DEPENDENT MONOOXYGENASE 1 (FMO1) in the cytosol (for more details of the biosynthetic pathways and their evolutionary origin, see Reference [107]). The authors also describe that Ca^2+^ regulates SA and NHP biosynthesis by modulating the activities of CPKs, CBP60a/CPB60g, and CAMTA1/2/3, which might be a link to the traveling Ca^2+^/ROS/electric waves (Figure 2). Analyses of local and systemic resistance responses, as well as metabolite profiles in various knockout mutant combinations, suggest that NHP is the traveling and active compound. Moreover, it is also involved in the local defense response. NHP can be further converted to NHP-OGlc by the GLYCOSYL TRANSFERASE 76B1 (UGZ6B1), and NHP-OGlc travels along with NHP [108,109,110]; however, mutant analyses showed that NHP and not the glycosylated form is active. Upon local pathogen attack, the expression of the *ALD1*, *SARD4* and *FMO1* genes is strongly stimulated which results in Pip and NHP accumulation [111,112,113]. Besides numerous defense-related transcription factors which are targeted by NHP both in local and systemic tissue, Guerra et al. [114] showed that the CALCIUM-DEPENDENT PROTEIN KINASE 5 (CPK5) is involved in NHP accumulation. NHP is linked to SA functions in systemic signaling and stimulates SA accumulation by activating the expression of SA biosynthesis genes. NHP also stimulates the expression of its own biosynthesis genes which leads to an amplification of the response. However, neither details about the traveling mechanism nor perception of NHP in local or systemic cells is known yet. Upon pathogen attack, NHP accumulates in leaf phloem sap of cucumber and Schnake et al. [115] proposed that the chemical compound is a phloem-mobile SAR signal. It is also worth mentioning that the biosynthesis genes and NHP itself is present in various mono- and dicots. Therefore, NHP is a good candidate for a generally used systemic signaling molecule [110,115] and it is linked to the traveling signaling compounds SA and Ca^2+^ (via CPK5) (Figure 2).

### 12.5. Dehydroabietinal (DA)

The tricyclic diterpenoid DA is synthesized from geranylgeranyl diphosphate by the sequential action of diterpene synthases and P450 monooxygenases. The abietane diterpenoid synthesis occurs in chloroplasts and starts with the cyclization of geranylgeranyl diphosphate by diterpene synthase. The final oxidation step is mediated by the abietane oxidase, a cytochrome P450 monooxygenase located in the cytosol [116,117]. DA induces SAR in picomolar concentrations in the leaves. The response requires SA and DIR1 in systemic tissue [94,103]. ^2^H-DA labeling experiments suggest that DA moves from the infection site to distal tissue. Furthermore, induction of SAR requires uptake DA from a low molecular weight faction into a trypsin-sensitive high molecular weight signaling form of about 100 kDa [94]. Interestingly, DIR1 was also found in the DA-containing high molecular weight fraction (>100 kDa) that is capable of inducing SAR. The relationship of DA and DIR1 in systemic signaling is not yet clear. The lipid-binding protein DIR1 [93,118] might be required for DA movement. It is also discussed that DA and DIR1 are part of a so-called “SAR signalosome”, that mediates long distance signaling. How lipid metabolites move in the aqueous phloem environment and how this movement can be coordinated is a general problem in systemic signaling. However, examples from animal systems suggest that binding to proteins is the most reasonable explanation (see Reference [118]).

DA also promotes flowering in *Arabidopsis*, and several flowering-related proteins are also linked to systemic signaling events and SAR (see References [103,119]). FLOWERING LOCUS D (FLD) promotes flowering by suppressing expression of the flowering repressor *FLOWERING LOCUS C* (*FLC*) through histone modifications. This results in the production of the flowering protein FLOWERING LOCUS T (FT). The FT protein moves from leaves to the shoot apical meristem through the phloem. DA and bacterial infections also stimulate *FLD* expression in treated and distal leaves. During flower induction, FLD is required for *FT* expression, whereas during induced SAR, FLD stimulates SA accumulation in distal leaves [120,121]. FLD is also required for the induction of SAR by AzA [120]. It appears that FLD is an important player in SAR and functions upstream of SA.

### 12.6. Defective in Induced Resistance1 (DIR1) 

The small 7 kDa lipid transfer protein DIR1 was identified in screens for mutants with SAR-deficient phenotype [122]. Additional studies uncovered that also other DIR1-like proteins, lipid transfer proteins (LTPs, such as LTR1 and LTR2) and AZI1 and its homologs have similar functions in model and crop plants (summarized in References [83,88,103,123,124]). Protein–protein interaction assays showed that DIR1 interacts with DIR1-like and other LTPs. Carella et al. [125] showed the importance of multiple LTPs for SAR in *Arabidopsis*. DIR1 is involved in the signaling functions together with AzA, DA, G3P and MeSA, and found in exudate preparations as a high molecular mass oligomeric form [94].

DIR1 is present in phloem sap-enriched petiole exudates suggesting that the protein gains access to the phloem for movement [126]. Carella et al. [127] investigated how DIR1 is transported to distant leaves during SAR. They used transgenic *Arabidopsis* lines with reduced cell-to-cell movement by overexpressing the PDLP1 and PDLP5. The overexpressor lines were defective for SAR, and DIR1 antibody signals were not observed in phloem sap-enriched petiole exudates collected from distant leaves [127]. The study clearly demonstrates that cell-to-cell movement of DIR1 through PD is important during long-distance SAR signaling in *Arabidopsis*.

### 12.7. eATP

Cytoplasmic ATP concentrations range between 0.5 and 1 mM in a cell and ATP and can be released into the apoplast by PM-localized transporters or exocytosis [128]. Removal of ATP from the apoplast causes cell death [129] and eATP has been long discussed as stress signaling molecule, since its level increases after wounding or pathogen attack. For instance, *P. syringae* infection results in increase in eATP levels [130] which cause local immune responses including [Ca^2+^]_cyt_ elevation, NO and ROS production, MAPK and defense gene activation [131]. eATP is perceived by the legume-like lectin receptor-like kinase I.9 (LecRK-I.9, or P2K1), a member of the purinoreceptor (P2K) family (also called DOES NOT REPSOND TO NUCLEOTIDES1, DORN1) [132]. Pham et al. [133] identified P2K2 a second receptor for eATP and both receptors have redundant functions. The Ca^2+^ channel(s) which are targeted by eATP signals are not known yet. Upon activation by eATP binding, P2K1 and -2 dimerize and phosphorylate each other. An immediate target, at least of P2K1, is RBOHD which generates apoplastic ROS after phosphorylation. Proposed downstream responses are stimulation of jasmonates and the activation of the CALMODULIN-BINDING TRANSCRIPTION ACTIVATOR3 (CAMTA3) [132,134,135], which, again, demonstrates links to other systemic signaling molecules (Figure 2). How far eATP can travel or diffuse is difficult to predict. eATP might be involved in transferring information to the immediate neighboring cells. However, it might also be a stimulating agent for amplifying or supporting long-distance traveling signals as part of a network that involved other long-distance signaling molecules, such as ROS and Ca^2+^ (Figure 2).

### 12.8. eNAD(P)

Also apoplastic NAD(P) stimulates immune responses, SAR and SA signaling including *PATHOGENESIS-RELATED* (*PR*) gene expression. Zhang et al. [136,137] used the human NAD(P)-metabolizing enzyme CD38 to manipulate extracellular NAD(P) levels in transgenic *Arabidopsis* and confirmed that the amount of apoplastic eNAD(P) correlated to the elicited local responses and SAR. eNAD(P) is perceived by the PM-localized NAD(P) receptors LecRK-I.8 and LecRK-VI.2 [138,139], which belong to the same protein family as P2K1/2 responsible for eATP recognition. LecRK-I.8 binds NAD but not NADP, whereas the extracellular lectin domain of LecRK-VI.2 binds NAD and NADP [139] (Figure 2). LecRK-VI.2 interacts with BRASSINOSTEROID INSENSITIVE1-ASSOCIATED KINASE1 (BAK1) and BAK1-LIKE1 (BKK1), which connects eNAD(P)-induced immune responses to those of other receptors involved in plant immunity [139]. Infection assays and mutant analyses showed that eNAD(P) confers local resistance against *P. syringae* [136] and participates in distal responses via SAR [136,137,138]. eNAD(P) might have similar functions in supporting traveling compounds. It is unlikely that eNAD(P) moves far away from the cell where it is produced, similar to eATP.

### 12.9. Rapid Alkalization Factors (RALFs)

Besides receptors for eATP and eNAD(P), FERONIA (FER) has been identified as receptor for RAPID ALCALIZATION FACTORs (RALFs). RALFs control the extracellular pH, proton fluxes across the PM, stimulate MAPKs and ROS accumulation. Similar to eATP and eNAD(P) signaling, the RALF–FER pathway starts with a rapid increase in [Ca^2+^]_cyt_ levels. It is involved in developmental and defense responses and linked to hormones such as jasmonates. Therefore, it is likely that the RALF–FER pathway also influences systemic signaling; however, barely any investigations have been performed so far. Extracellular pH changes have tremendous effects not only on local responses but influences also many aspects of signal propagations, through control of apoplastic transport, transport across the PM and distribution of systemic signaling compounds. The pathway might participate in the propagation of Ca^2+^/ROS/electric waves or distribution of other long-distance traveling compounds which have to cross the PM (Figure 2).

## 13. Volatiles or Gaseous Compounds

Although long known, volatiles became important novel long-distance information carrier with the ability to interfere with traveling signals at any place along the traveling path. Furthermore, it is long known that systemic signals can only travel from one leaf to another when they have direct vascular connections (see Reference [47] and references therein). Information transfer via volatiles does not require such a connection, as long as the receiving cells/tissues can perceive them. Volatiles also offer the possibility to transfer information from one plant to neighboring plants or within the entire community [140,141]. Therefore, they play important roles for shaping plant communities and are the result of a long co-evolution of the community members, or interacting organisms (for ecological details, see References [140,141,142,143,144,145,146,147,148,149,150]). Volatiles belong to various chemical classes, such as terpenoids, benzenoids, phenylpropanoids and fatty acid-derived molecules, as well as minor classes such as nitriles, (ald)oximes and sulfides [140]. The gaseous compounds include also the hormone ethylene, the signaling compound NO, but also MeSA (cf. above). MeSA can travel through the phloem but is also released into the air and a similar propagation mechanism is discussed for NO [84]. For more information about ethylene and NO, I refer to more specialized reviews on their functions (e.g., Reference [84]).

Emission of volatiles in response to damage or other types of threat is well known. Examples are green leaf volatiles [148,149] (such as (*Z*)-3-hexenal, (*E*)-2-hexenal, (*Z*)-3-hexen-1-ol or (*Z*)-3-hexen-1-yl-acetate) deriving from the oxylipin biosynthesis pathway, terpenes from the plastid-localized methylerythriol-phosphate and the cytosolic mevalonate pathways and aromatic phenylpropanoids and benzenoids from the shikimate pathway, which also includes MeSA. For none of the volatiles (except ethylene) a perception system is known at the molecular level yet. However, as recently summarized in [149] the defense-related volatiles trigger downstream responses, such as [Ca^2+^]_cyt_ elevation; ROS production; MAP-kinase activation; defense gene expression, including genes for jasmonate and SA biosynthesis and signaling. The responses are identical in local and systemic tissues, but also along the traveling path. Therefore, it is reasonable to assume that volatiles may have two functions: They can support long distance signaling by stimulating or amplifying a traveling process along its path in the plant tissue, e.g., the phloem, and they might activate defense responses de novo in distal tissues, for instance in a neighboring plant. Heil and Karban [150] observed that bigger and anatomically more complex organisms prefer volatiles for the distribution of threat information whereas signaling through the vascular system is more often used for smaller and less complex plants.

Erb et al. [151,152] showed that the herbivory-induced aromatic compound indole might be a potent priming agent in *Zea mays* which increased the accumulation of defense-related phytohormones and volatiles in undamaged neighboring plants. In spite of some species-specific differences, the authors also showed that synthetic indole triggered the emission of other volatile compounds including α-pinene, and (*E*)-β-caryophyllene in cotton and cowpea. Apparently, there might exist a hierarchy of volatiles with indole being an upstream component which induces emission of further downstream volatiles such as α-pinene. The rapid progress in volatile detection systems already started to change our understanding of systemic signal propagation. Volatiles from many origins [151] interfere with the plant defense machinery. Since also many microbes, in particular those forming beneficial symbiotic interactions with roots, emit volatiles which can be perceived by the aerial parts of the plant, signal propagation via volatiles might be an additional long overlocked way for the distribution of threat information and activating the immune system in distal parts of the plants. Besides the ecological importance, the ability to apply volatile compounds to a particular plant tissue is also a powerful tool to understand how they interact with other signaling molecules traveling within the plant body. The response to volatile applications might help to understand, how the tissue-bound and gaseous signal propagation mechanisms are connected and how regulatory feedback and amplification processes—which are currently discussed for systemic signaling compounds traveling through the vascular tissue—might be activated from the outside. Considering the requirement for a fast distribution of threat information, these two transmission pathways will probably not compete but complement each other.

## 14. Monoterpenes

Riedlmeier et al. [152] identified monoterpenes as volatile compounds involved in SAR. A mixture of the bicyclic monoterpenes α-pinene and β-pinene induced defense, accumulation of ROS, and expression of SA- and SAR-related genes, including the SAR regulatory *AZI1* gene and its paralogs. The monoterpenes acted in parallel with SA. Not surprisingly, the volatiles induced also defense in neighboring plants; therefore, they function as infochemicals in plant-to-plant signaling and allow defense-signal propagation between neighboring plants and ultimately in plant communities. Wenig et al. [153] extended the studies and presented a model, in which a SAR-inducing infection triggers Pip and G3P accumulation which stimulate each other in a positive feedback loop. This results in monoterpene synthesis and emission with the downstream targets LLP1 and AZI1. At the side of perception in the systemic tissue of the same plant or the neighboring plant, the volatile activates LLP1, followed by AZI1 and downstream immune responses. LLP1 not only signals downstream to activate the SAR immune response but also drives a feedback loop with Pip and G3P to further stimulate monoterpene biosynthesis and emission [153]. Volatile compounds are probably very helpful tools to decipher the components which transfer the information to distal tissue. More importantly, these volatiles might not only activate genes for the local or systemic responses, but also those for signal propagation. The higher flexibility of volatiles to induce defense responses and to support existing traveling processes within the plant body makes them important players in systemic signal propagation processes. Considering the increasing number of volatiles involved in information exchange within and between organisms, the identified monoterpenes are probably only the peak of the iceberg in the identification of gaseous compounds as information carriers.

## 15. How Do These Chemical Compounds Interact?

The literature survey shows numerous links between the signaling compounds (Appendix A) and many of them might be part of a signal propagation network (with proposed feedback loops and amplification processes) (Figure 2), but which of these components are actually traveling, which of them initiate the traveling process at the local side but do not travel themselves, and which of them realize the defense program in the distal tissue in response to a traveling signal, is not completely resolved. SA as hormone travels in the phloem, MeSA is present in the phloem but also volatile. Moreover, Pip and NHP appear to be mobile and NHP is found in the phloem sap [115]. Wang et al. [154] demonstrated that Pip confers SAR by increasing levels of the free radicals, NO, and ROS, which act upstream of G3P. Plants defective in NO, ROS, G3P or SA biosyntheses accumulate reduced Pip in their distal uninfected tissues, although they contain wild-type-like levels of Pip in their infected leaves. These data indicate that de novo synthesis of Pip in distal tissues is dependent on SA, G3P, ROS and the gas NO (see Reference [154]). In contrast, the C_9_ dicarboxylic acid AzA accumulates downstream of NO and H_2_O_2_ but upstream of G3P [155,156,157]. G3P is proposed to be part of a positive feedback loop which includes the lipid transfer proteins AZI1 and DIR1 [93,123]. Additional participants in the signaling process are the diterpenoid DA and the volatile monoterpenes α- and β-pinene [152,153]. All these compounds are controlled by feedback loops and amplification processes, but whether they operate during traveling or only at the target tissue, is not clear. This already indicates that the compounds cannot be simply classified as “inducers”, “travelers” and “realizers”. It may probably be helpful to distinguish between those compounds (i) which travel and activate a response program only in a distant tissue and those (ii) which permanently activate their de novo synthesis along the traveling path. Furthermore, a clear picture on compounds (i) which travel and activate downstream responses on their road, and those (ii) which travel as inactive forms and become only activated and, thus, active in the systemic tissue, is missing. For compounds which travel as active forms, it remains to be determined whether they activate the responses along their traveling path or only in the systemic tissue. If regulatory loops, amplification processes and feedback mechanisms come into play, it should be clarified whether they are required for the movement of the systemic signal or only for the realization of the response in the distal tissue. Finally, volatiles can activate the response in the distal tissues. For ethylene, only cells can respond which express the ethylene receptor ETR1. Barely anything is known about volatile perception in systemic signal propagation. Are the volatiles also required for the propagation of the information within the tissue, or – after release from the infection site – only required for realization of the response in the distal tissue? It would also be helpful to define a responding systemic tissue. A distant leaf with a vascular connection to a threat-exposed leaf can respond to systemic signals traveling through the vascular system, whereas a leaf without such a direct connection cannot respond to these signals, but responds to volatiles. In molecular terms, the decoding process of the traveling information must be different.

Recently, Kachroo and Kachroo [85] summarized the genetic, molecular and biochemical studies and proposed that the “SAR inducers operate in a bifurate pathway”: an SA regulating pathway which is fed by DA, NHP and eNAD(P); and a pathway consisting of Pip, G3P, NO, ROS and AzA. However, Pip is a precursor of NHP, and, therefore, a link between these systemic signals is expected. Wenig et al. [153] extends the concept to monoterpenes. According to their model, SAR triggers Pip and G3P accumulation, which stimulate each other via a positive feedback loop. This results in monoterpene emission. The volatile activates the downstream SAR compounds LLP1 and AZI1, either along the propagating path within a plant, or by the generation of a wave of immune signaling within canopies or plant stands.

The involvement of volatile compounds (such as NO, ROS or monoterpenes) should allow to clarify which of the proposed compounds which propagate in the plant tissue are induced in the systemic tissues but do not necessary travel in the plant. It is also possible that maintenance of signal propagation by traveling compounds within the plant body requires a permanent refreshment along the path, and that requires the presence of volatiles. This can be tested, e.g., by comparing volatile concentrations with the concentrations of the other systemic signaling compounds in the plant tissue along the traveling path and in the terminal systemic tissue. Furthermore, since volatiles reach tissues which are not directly connected through the vascular tissue, a comparative analysis could clarify whether volatile-mediated systemic responses and responses mediated by traveling compounds within the plant body operate independently or complement each other. Since barely anything is known about the perception of volatiles in plants, it is still difficult to understand how they stimulate the accumulation of the systemic compounds in their target tissues.

## 16. Traveling RNAs

A huge number of RNA travel within a plant and between different species, and they are important players in developmental processes and immunity. The importance and dimension of traveling RNAs make it unlikely that information transfer via RNAs operates independently of that of the other traveling compounds. However, barely anything is known about possible connections.

As information-carrying molecules, various types of RNAs (messenger RNAs (mRNAs), small interfering RNAs (siRNAs) and microRNAs (miRNAs), but also ribosomal and transfer RNAs (rRNAs and tRNAs)) can move locally between cells through PD, and systemically over long distances through the phloem [158,159,160,161,162,163]. Besides these two mechanisms, RNA trafficking between cells occurs also via vesicles (exosomes) [164]. The non-cell-autonomous RNAs regulate numerous developmental processes, and recent studies showed that they also play important roles in systemic defense responses and protection (see Reference [165]). Many examples are nicely described in a mechanistical context by Li and Chen [162]. Well-investigated mobile RNAs moving through the PD from one cell to the neighboring cell are the transcription factor *KNOTTED1* mRNA [166], the sucrose transporter *SUC1* mRNA [167], as well as the *miR390* [159] and *miR165*/*166* [161]. Long-distance transport through the phloem has been proposed for many RNA species, which were identified in the phloem of tissues where they were not generated [168,169,170]. These observations are often based on grafting experiments. For instance, Zhang et al. [171] showed that expression of a tomato prosystemin gene in *Arabidopsis* reveals systemic translocation of its mRNA and confers necrotrophic fungal resistance.

The PD and their modified structures, the PD pore units, participate in short distance cell-to-cell transport as well as phloem loading. Local transport occurs via two or more neighboring cells, which are connected by PD. For long-distance transport through the phloem, the RNAs are transported from the companion cells to the living enucleated sieve elements (which lack the transcription machinery). Both cell types are connected via the PD pore units. Long-distance RNA transport through phloem is possible because the phloem sap does not contain RNase activity [172,173] (Figure 3).

The PD and the PD pore units contain proteins and RNAs. Yoo et al. [174] showed that the pumpkin Phloem Small-RNA Binding Protein1 (CmPSRP1) can bind single-stranded small RNAs which move through the connections between companion cells and sieve elements and, thus, it participates in the loading of the phloem sap with RNAs [175]. Similarly, the mRNA for the sucrose transporter SUC1 must be translocated to the sieve elements via PD pore units [167] since the *suc1* mRNA is found in companion cells and the enucleated sieve elements. The tomato *prosystemin* mRNA moves through the phloem and the graft interface and is unloaded to nucleated cells of the scion, where it is translated in response to herbivore and pathogen attack [171]. The mobile cyclophilin *SlCyp1* mRNA controls the root-shoot ratio and the mobile antiflorigen *PEBP* mRNA inhibits flowering in tomato–tobacco heterografts [176]. More than 3500 mobile RNAs have been described since then that move from shoots to roots, by grafting experiments both with model organisms like *Arabidopsis* [170] or agriculturally important crops [177], and many of them are involved in plant defense.

Besides mRNAs, numerous small RNAs were identified in the phloem sap. These small RNAs, consisting of 21–24 nucleotides, are miRNAs and siRNAs that mediate RNA silencing. miRNA genes are transcribed by RNA polymerase II which are cleaved by DICER-LIKE1 from their precursor RNAs to produce mature miRNAs. They form the RNA-induced silencing complex (RISC) with ARGONAUTE1, which mediates cleavage of target RNAs or activates their translational repression [178].

siRNAs derive from double-stranded RNAs produced by the RNA-dependent RNA polymerases or from the transcription of inverted-repeat sequences. Besides posttranscriptional control on mRNA levels in the cytoplasm, small RNAs can also mediate transcriptional gene silencing through DNA methylation or histone modifications in the nucleus [179].

The presence of small RNAs in the phloem sap clearly demonstrates that they are mobile and, thus, involved in long-distance signaling. *miR399* is a long-distance signal for the regulation of plant phosphate homeostasis [180]. Under nutrient starvation conditions the *miR395*, *miR398*, and *miR399* levels increase in the phloem sap of *B. napus. miR827* and *miR2111* are also traveling over long-distances, while their respective mRNAs do not travel [181]. In Legumes, *miR2111* is translocated from shoot to root, to regulate rhizobial infection [182]. On viral infection, siRNAs may act as mobile signals and move faster than the viral spread to prime antiviral gene silencing in not yet infected cells [183,184]. These few examples highlight the potential of traveling small RNA species for systemic defense responses and it is likely that many more examples will be discovered.

Exosomes are vesicles of about 30–100 nm in diameter which are released from a cell into the apoplast and taken up by neighboring cells (Figure 3; for more details, see Reference [185]). Besides proteins, various types of RNAs can be transported between cells via exosomes. Therefore, exosomes represent an important intercellular communication system and are well investigated in animal cells (see Reference [185]). A number of siRNAs and miRNAs are transferred from *Arabidopsis* to *Botrytis cinerea* through extracellular vesicles to silence fungal virulence genes and contribute to host immunity (cf. below). This suggests the existence of an exosome-mediated secretion pathway in plant–fungal pathogen interactions [186]. Taken together, the combination of the three possible transportation systems for RNAs (PD, exosomes and phloem) allow the distribution of signaling molecules in the entire plant body, and even beyond this into interacting other species. It will be important to investigate which of the traveling compounds also utilize the exosome passage and how this is controlled.

### Inter-Organismic RNA Transfer

RNAs can travel between hosts and their interacting microbes. Hailing Jin’s group showed that the necrotrophic fungal pathogen *B. cinerea* produces small RNAs during infection that hijack the host plant’s RNAi machinery to silence genes of *Arabidopsis* and tomato involved in host immunity [187]. Shahid et al. [188] demonstrated that micro RNAs from the parasitic plant *Cuscuta campestris* target host mRNAs and induce the production of secondary siRNAs which degrade host defense mRNAs. The trafficking of small RNAs is bidirectional; plants can also deliver endogenous small RNAs into invading fungal pathogens. *Arabidopsis* siRNAs and miRNAs have been identified in the associated pathogen *B. cinerea* and contribute to host immunity. Furthermore, *miR166* and *miR159* generated in cotton have been shown to be transferred to the hyphae of the wilt pathogen *Verticillium dahliae* during infection, where they reduced expression of genes encoding a Ca^2+^-dependent cysteine protease (*Clp-1*) and an isotrichodermin C-15 hydroxylase (*HiC-15*). Deletion of those two genes in the fungus inhibited microsclerotia formation or hyphae growth, respectively, and down-regulation of *Clp-1* and *HiC-15* through small RNAs from the host plant interfered with fungal pathogenicity [189]. Ye et al. [190] showed that *Erwinia chrysanthemi*, a destructive pathogen, causes soft-rot diseases in orchid leaves. The pathogen restricts the accumulation of *NBS-LRR Resistance* mRNAs by stimulation the abundance of their target miRNAs *miR524650*, *miR1510a**, *miR2118* and *miR5246*. Colonisation of the roots by the beneficial fungus *Piriformospora indica* reversed this regulation in the leaves and confers better resistance against *E. chrysanthemi*. This suggests that either signals or miRNAs travel from the roots to the leaves. About half of the transcriptome of the parasite *Cuscuta*, growing on an *Arabidopsis* plant, is of *Arabidopsis* origin [191]. This demonstrates that thousands of mobile transcripts migrate across the parasite’s haustorium from plants to *Cuscuta* species [191,192,193,194]. A growing number of recent studies suggest that both plants and fungi use these cross-kingdom RNAi strategies [189,195,196,197,198,199,200,201,202,203]. For both plants and fungi, vesicles containing small RNAs are secreted to the extracellular space and taken up by the partners. Since the RNA species travel through extracellular vesicles and silence fungal virulence-related genes, they might as well be used as a strategy to control pests in agriculture. Host-induced gene silencing provides a disease control alternative to chemical control and breeding. Again, the question arises whether these vesicles also transport other molecules involved in systemic defense regulation.

But how and why do RNAs move and how is this controlled? The small RNAs can be transferred nakedly, either bound by proteins or packed into vesicles. All these forms could move passively. Alternatively, an active movement of selective RNA species requires *cis*-regulatory elements or secondary structures that can be recognized by any transportation machinery. One possibility is selective binding to proteins. Another possibility could be regulatory processes that control trafficking of the RNA or RNA/protein complexes through the PD. For long-distance transport, this could determine entry into the phloem sap. Callose deposition is probably one mechanism to regulate the width of the PD. Lim et al. [92] compared the transport of the signaling molecules AzA, G3P, and SA and showed that AzA and G3P transport occurs via the symplastic route and PD while SA moves via the extracytosolic apoplast. The PDLP 1 and -5 are required for SAR even though permeability through the PD in *pdlp1* and -*5* mutants was comparable to that of wild-type plants. Overexpression of PDLP5 drastically reduced PD permeability and impaired SAR. PDLP1 interacted with AZI1. The lipid transfer-like protein required for AzA- and G3P-induced SAR, cf. above) and contributed to its intracellular partitioning. These results highlight the regulatory role of PD-localizing proteins in SAR, and simultaneously demonstrates that these finding are probably only the pike of the iceberg [92]. It is important to investigate whether PD permeability affects all signaling molecules. Gating of the PD has already been shown to control AzA and G3P translocation [96].

Overall, the rapid increase in our knowledge about traveling RNAs opens the question, how they are connected to the other above described systemic signaling systems. It is difficult to believe that an RNA-based information transfer within the plant operates independently of all the other traveling compounds described above. It is tempting to speculate that RNAs are part of huge network of multiple traveling compounds. This may help to understand how a systemic response can be specific to a threat perceived at another end of the plant.

## 17. More Questions Than Answers

The main questions that need to be addressed are as follows: Which compounds travel and which are only part of the systemic signaling process, but do not move, i.e., initiate the traveling process in the local tissue/cell or decode the information in the systemic tissue/cell? How is a specific answer in the systemic tissue achieved? Information transfer via RNAs allows highly specific information transfer. However, considering the huge amounts of RNAs which (can) move in(to) the vascular tissue, a specific response in the distal tissue requires a decision which of the RNAs travel. Furthermore, are the signals travel independently of each other or is their movement coordinated? Finally, there must be crosstalks between the traveling signals. While some crosslinks are likely (Ca^2+^, ROS, hormones, volatiles, etc.), barely any information is available about crosstalks between mobile SAR signals and traveling RNAs. At least loading of the phloem appears to be a common mechanism for these systemic signals. Which mechanisms control entry into the vascular tissue? Are they the same for traveling RNAs and metabolites? 

How is specificity achieved? While some of the traveling compounds per se, do not carry specific information (e.g., Ca^2+^), local threats induce threat-specific Ca^2+^ signatures, which induce appropriate responses. In combination with ROS, electric and hydraulic signals, an even higher level of specificity can be achieved. However, it is not clear whether the local signatures are maintained during traveling. Finally, a systemic signal that transfers only alarm information might induce a general broad-spectrum response while other systemic signals may carry threat-specific information that induce the same or similar responses in the local and systemic tissues. There must be a gradient from less-specific to highly specific responses in the systemic tissue. Systemic signals with less information might have the advantage that they allow cross-tolerance responses and are easier to store.

The term “systemic signal” covers a broad spectrum of information transfer within a plant and a community. It is probably helpful to distinguish between those information systems which operate only in their near environment (cell-to-cell transfer via PD or the apoplast), those which release compounds into the vascular system and, thus, allow information transfer over longer distances, and those which utilize gaseous compounds and are therefore not bound to plant organs. Clearly, there are connections, but the molecular basis differs.

Future investigations of systemic signaling in plants require a multidisciplinary approach that brings together cellular, physiological, molecular and ecological research. This will allow us to understand communication within a plant and its environment.

## Figures and Tables

**Figure 1 ijms-22-03152-f001:**
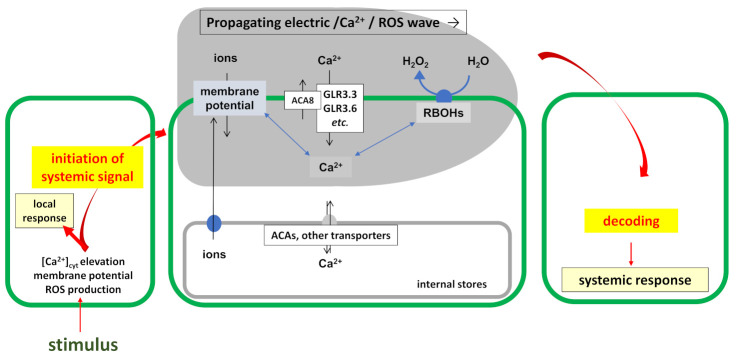
The model describes a propagating electric/Ca^2+^/ROS wave (middle cell), its initiation in a local cell (left cell) and its decoding in a receiving cell (right cell). The participating Ca^2+^-im- and exporters (GLR3.3, -3.6, ACA8 and other ACAs) and the H_2_O_2_-producing NADPH oxidases (RBOHs) are described in the text. Internal stores: membrane-surrounded cell-internal compartments.

**Figure 2 ijms-22-03152-f002:**
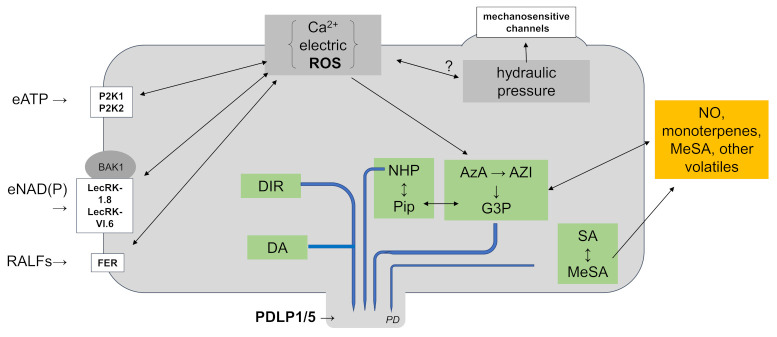
Potential interactions of signaling molecules in a cell involved in systemic signal propagation. The errors (↔) refer to interactions described in the text. The blue lines indicate transport of signaling components through plasmodesmata (PD). Characterized channels and receptors are in white boxes. The brackets around “ROS, Ca^2+^ and electric” symbolize a wave. The “?” refers to an unclear connection or physiological relevance. AzA, azelaic acid; AZI, AzA-INDUCED1; G3P, glycerol-3-phosphate; DA, dehydroabietinal; DIR, DEFECTIVE IN INDUCED RESISTANCE1; Pip, pipecolic acid; NHP, N-hydroxy-Pip; SA, salicylic acid; MeSA, methyl-SA; PDLP1/5, PD-localized proteins 1 and 5.

**Figure 3 ijms-22-03152-f003:**
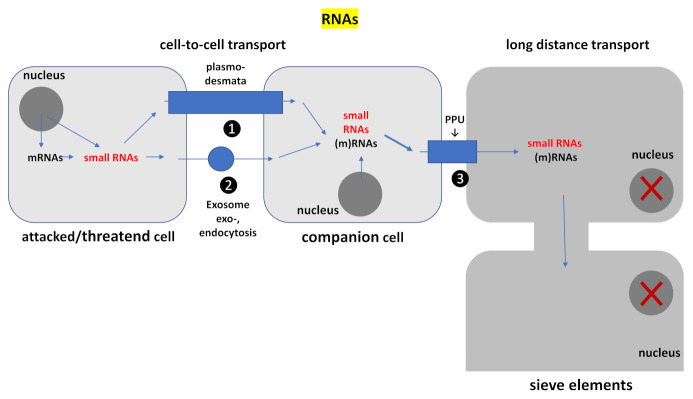
RNA transport through plasmodesmata (1), exo- and endocytosis (2) and plasmodesmata pore units (PPU) (3) from attacked/threated cells via the companion cells to the sieve elements. The small RNAs, consisting of 21–24 nucleotides, are miRNAs and siRNAs that mediate RNA silencing.

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
