# Peer review of "Threat at One End of the Plant: What Travels to Inform the Other Parts?"

_ijms, 2021, doi:10.3390/ijms22063152_

Round 1

Reviewer 1 Report

The review by Ralf Oelmueller discussed in length and detail systemic signaling in plants and also branches out to cover related aspects of signaling between individual plants. Overall, this review will be a tremendous source of information for the research community. However, I have few comments to improve the review, mostly targeted at making the text more accessible to non-expert readers.

  1. The review is very long. Now with the complexity of the topic I doubt that the material covered could be covered in shorter space. However, I would suggest the use of more different headers to better clarify sections and subsections, perhaps during final formatting. I also would suggest to break the text up into individual paragraphs even more than at the current state.
  2. Lines 83-84: The authors describes the upregulation of PRRs and MAP kinase cascade components. It should be clarified whether this refers to higher transcript abundance or higher protein levels.
  3. Line 87: "interested reports" should be "interesting reports".
  4. In the paragraph starting at line 103, the author discusses that previous exposure to a stimulus can result in higher activation of defense responses. However, there are also examples, where application of a novel stimulus within a short time-window will not result in a response, for example stomatal closure in response to ozone (see Vahisalu et al., 2010 https://onlinelibrary.wiley.com/doi/full/10.1111/j.1365-313X.2010.04159.x). This is not contradictory with the topic the author discusses but should be included/mentioned.
  5. ROS: The author mentions that RBOH activity is regulated by phosphorylation of the N-terminal region and mentions just one example of regulation via the C-terminal region by S-nitrosylation. However, recent reports describe that the C-terminus of RBOHD is also a target for regulation by phosphorylation (Kimura et al., 2020 http://www.plantcell.org/content/32/4/1063; Lee et al., 2020 https://www.nature.com/articles/s41467-020-15601-5) but also persulfidation (Shen et al., 2020 http://www.plantcell.org/content/32/4/1000). Furthermore, the arguably best understood example of a protein kinase regulation RBOHD, BIK1, is not mentioned in the context. In addition, the strong linking of ROS with calcium is still poorly understood, but it is technically possible to activation RBOH-dependent ROS production in the absence of calcium, and vice versa.
  6. Calcium: Some of the recent work identifying calcium channels involved in immunity has not been included (Thor et al., 2020 https://www.nature.com/articles/s41586-020-2702-1; Tian et al., 2019 https://www.nature.com/articles/s41586-019-1413-y). This would be especially important, since the kinase BIK1 is involved in controlling RBOHD-dependent ROS production but also OSCA-dependent calcium fluxes. Furthermore, the OSCA and CNGC classes of calcium channels should be described. CNGC is used later in the review but is not explained.
  7. Line 335: The statement that CLE peptide signaling elicits ABA production is referenced with four review articles. It would be better to reference the original work, possible together with the reviews.
  8. Reference 50 (Lee et al) is missing the author last name in the reference list. I suggest careful checking of all references in the manuscript. References from 131 onwards also are double numbered in the list.
  9. Lines 418: Pipecolic acid has already been expanded earlier. On the other hand pipene is not explained and G3P is not expanded at this point.
  10. Sa derivative MeSA: for a long section discussing MeSA as a community messenger there are no references. Is there original research to support the section? If not this should be more clearly highlighted as speculation. Especially considering the paragraph at the end of the same page.
  11. Glycerol-3-phosphate: Here the author uses quotes to use direct phrases from other articles. While acceptable this can be confusing. I would suggest to rephrase the statements to integrate them better with the manuscript. The same suggestion applies for later uses of direct quotes.
  12. Lines 516 onwards: The fonts used for the manuscript are off.
  13. Figures are misnumbered: Figure 3 comes before Figure 2, which is never mentioned in the text but rather another figure is also labelled as "Figure 3". This sould be corrected.
  14. Figure 2: The figure is very hard to understand. Blue lines are mentioned but at least in the review version some lines appeared more grey. Does this mean anything? I would suggest to find another, less complicated way to draw the figure which makes it easier for the reader to use the massive amount of information in the figure. Perhaps breaking it up into two individual panels could be helpful.
  15. Line 673: "SA regulating pathway which if fed..." should likely be "SA regulating pathway which is fed...".
  16. Lines 685-686: The author states that "This can be tested quite easily for systemic responses involving volatiles". I would recommend to give more precise examples of how this could be tested easily, especially since I could see this directly in the subsequent section.
  17. Lines 730: Are there references for the ecological importance of volatiles as signals?
  18. The section on SAR, Pip, G3P, LLP1 and AZI1 is hard to understand and potentially misleading. I would recommend to rephrase and simpligy this for clarity for the non-expert reader.
  19. Line 828: The reader is directed to Figure 4 for more details. However, there is not more detail in figure 4.
  20. Overall, the figures could be revised for appearance and for clarity.

Author Response

i

Reviewer 2 Report

The review paper “Threat at the one end of the plant: What travels to inform the other parts?” is desirable for plant research community. The title catches the reader’s attention and the abstract outlines the key points of the paper. In general, the manuscript is well-organized, and the authors also summarized some compelling figures. However, the manuscript have several drawbacks about some details. Please check my comment below.

  1. The terminology for compounds is confusing. For example, Line 11. hormones and volatiles belong to small molecules (in metabolomics, metabolites = small molecules), thus it’s uncommon to list them as separate entities.
  2. All abbreviations used in figures should be defined in the figure caption, even though the abbreviations are defined in the text. This is helpful to read and understand the figures without digging into the text.
  3. Line 122-130, the authors should summarize the cited literature in less words instead of directly taking sentences from the literature.
  4. Line 269-272, the two single-sentence paragraphs need to be better integrated into the previous passages.
  5. Line 283-285, it’s hard to understand “deletion of one of the postulated compounds prevents also traveling of the other compounds”. Could this be clarified? How could a compound be deleted? Which compound is referred to here?
  6. Line 550, This figure should be # 4 not #2. In the green circle, AzA appear twice. Also, G3P, AzA, and DA are small molecules, so they don’t represent members of a higher molecular weight complex. The author may want to rearrange this figure.
  7. The section of ‘How do these chemical compounds interact?’ starts in line 654 should come after the section ‘Volatiles or gaseous compounds’ starts in line 687, because volatile compounds are still part of chemical compounds. It would be good to include examples of the interactions of volatile compounds with other compounds if there are.
  8. The font size in Figure 1 and Figure 4 could be larger, since there are a lot of blank space around. Also, the order of 1 exo- and endocytosis and 2 plasmodesma are not consistent in the Figure and caption.
